# Eliminating Luck and Chance in the Reactivation Process: A Systematic and Quantitative Study of the Thermal Reactivation of Activated Carbons

Karthik Rathinam [1], Volker Mauer [2], Christian Bläker [2], Christoph Pasel [2], Lucas Landwehrkamp [1,*], Dieter Bathen [2,3] and Stefan Panglisch [1]

1   Chair for Mechanical Process Engineering and Water Technology, University of Duisburg-Essen, 47057 Duisburg, Germany
2   Chair for Thermal Process Engineering, University of Duisburg-Essen, 47057 Duisburg, Germany
3   Institute for Environment & Energy, Technology and Analytics e.V., 47229 Duisburg, Germany
*   Correspondence: lucas.landwehrkamp@uni-due.de

**Abstract:** Increasing environmental concerns, stricter legal requirements, and a wide range of industrial applications have led to growing demand for activated carbon worldwide. The energy-intensive production of fresh activated carbon causes significant $CO_2$ emissions and contributes to global competition for renewable carbon-based raw materials. Although (thermal) reactivation of spent activated carbon can drastically reduce the demand for fresh material, the reactivation process itself is still mostly based on experience and empirical knowledge locked into activated carbon companies. Despite the vast number of papers published in the field, practically relevant, systematic, and quantitative knowledge on the thermal reactivation process is barely available. This paper presents a simple and robust methodology for the development of a predictive model for the production of reactivated carbon with a defined product quality under energetically optimized conditions. An exhausted activated carbon sample was subjected to 26 reactivation experiments in a specially designed laboratory rotary kiln, whereas the experiments were planned and evaluated with statistical design of experiments. The influence of the reactivation conditions (heating rate, heating time, $H_2O/N_2$ volume ratio, and $CO_2/N_2$ volume ratio) on the specific surface area, energy consumption, yield, and adsorption capacity for diatrizoic acid were evaluated. The BET surface of the reactivated carbons ranged between 590 $m^2/g$ and 769 $m^2/g$, whereas the respective fresh carbon had a BET surface of 843 $m^2/g$. The adsorption capacity for diatrizoic acid measured as the maximum solid phase concentration $q_m$ derived from the Langmuir equation varied between 24.4 g/kg and 69.7 g/kg (fresh carbon: 59.6 g/kg). It was possible to describe the dependency of the quality criteria on different reactivation parameters using mathematical expressions, whereas the response surface methodology with nonlinear regression was applied to build the models. A reactivation experiment under statistically optimized conditions resulted in energy savings up to 65%, whereas the properties of the reactivated sample were close to the predicted values.

**Keywords:** water treatment; activated carbon reactivation; $CO_2$ saving; modeling; response surface methodology; design of experiments

## 1. Introduction

As adsorbents, granular activated carbon (GAC) or powdered activated carbon (PAC) from fossil carbon, nut shells, wood, and peat (to a limited extent) play a central role in drinking and wastewater treatment [1]. Nowadays, owing to stricter legal requirements for organic trace substances, an additional demand for fresh activated carbon (AC) in drinking and wastewater treatment facilities is to be anticipated [2–4]. At the same time, a steep increase in demand for fresh AC in the exporting countries may lead to a critical development in the available quantity and price [5]. Therefore, it has been widely suggested

to reactivate and reuse exhausted AC. Most importantly, reactivated carbons (RACs) are not only cheaper than fresh AC, but they also have a significantly better energy and $CO_2$ balance [6]. In addition, reactivation of exhausted AC reduces the dependency on raw materials and minimizes waste disposal as well.

Due to the financial and environmental advantages of RAC, many regeneration approaches such as microwave reactivation, electrochemical processes, acid-washing, thermal reactivation, and supercritical fluid regeneration have been proposed [7,8]. Nevertheless, thermal reactivation (ex situ) is still the method of choice in most cases [7,9]. This method is essentially based on heating the carbon material between 700 °C and 950 °C under a defined gas atmosphere, in which the previously adsorbed pollutants are desorbed and/or pyrolytically/oxidatively converted.

The restoration of the adsorption capacity during thermal reactivation occurs in several steps including drying, desorption, pyrolysis, and oxidation. Generally, the term "adsorption capacity" refers to the amount of adsorbate (e.g., organic pollutant) adsorbed per unit mass of the adsorbent. In exhausted activated carbon, the available adsorption sites within the material are to a large extend saturated with adsorbate or no longer accessible. Whereas the drying step in thermal reactivation (up to 200 °C) is accounted for by the (partial) removal of volatile adsorptives due to the resulting water vapor, the desorption step (200–500 °C) is accounted for by the removal of volatile and unstable compounds. The decomposition of non-volatile compounds takes place during the pyrolysis step (500–700 °C). As a result of the recombination reactions of radical fragments with the surface, blocking of the pore system and a reduction in the adsorption capacity might also occur. During the oxidation step (at above 700 °C), gases (e.g., water vapor, carbon dioxide, or air) are added to oxidize the remaining adsorbed molecules. The gases must be added sub-stoichiometrically to avoid quantitative decomposition of the AC [10]. The addition of carbon dioxide leads to a pore opening at a later stage of the activation process, which involves widening of the narrow microporosity, whereas the addition of water vapor widens the microporosity from the early stage of the activation process [11]. Despite being widely considered one of the most efficient processes, thermal reactivation suffers from several drawbacks such as incomplete desorption or pyrolysis due to an insufficient reactivation temperature, overactivation, and/or mass loss (low yield) due to unfavorable conditions, an undesirable reduction in particle size, or fouling (blocking of the pores) [12–14]. Further, reactivation under non-optimized conditions leads to excess energy consumption, making the process economically and ecologically less efficient.

The reactivation conditions have been reported to have a crucial influence on the quality of RAC and, thus, on their further use in water treatment [15,16]. Xin-hui et al. thermally reactivated exhausted AC acquired from the silicon industry by varying the temperature, steam flow rate, and reactivation time with a fixed heating rate. It was observed that among these three factors, the steam flow rate had the most significant influence on the iodine number of the RAC and moreover led to a faster uptake of methylene blue [9]. Landwehrkamp et al. reactivated an exhausted AC sample in three different ways in a full-scale reactivation furnace and assessed the RAC's ability to remove organic micropollutants. The adsorption capacity of the RACs was assessed in terms of a fixed-bed filter breakthrough curve for diatrizoic acid at environmental concentrations (ng/L), as well as in batch adsorption tests at different mg/L concentration ranges for different organic compounds. They achieved the highest adsorption capacity with an oxidative reactivation in a rotary kiln at 920 °C [17]. Cevallos Toledo et al. [18] examined the influence of chemical treatment, reactivation time, and reactivation temperature on the gold–cyanide adsorption capacity and the mechanical stability of AC. They achieved the best reactivation results via acid washing with $HNO_3$, and a subsequent thermal reactivation at 850 °C over 1 h. Although a comparatively high number of experiments was conducted, the authors did not apply any statistical tools to detect the interdependencies between the examined influence factors.

Despite various empirical studies, systematic and quantitative knowledge of the AC reactivation process itself is still missing. Furthermore, the quantitative influence of single reactivation parameters on the product quality is largely unknown. On the one hand, furnaces at AC manufacturing sites are generally too large and too expensive in operation to be available for systematic investigations. On the other hand, scientific studies often rely on ovens at a laboratory scale using only a few grams or even milligrams of material, thereby obtaining results of little practical relevance. Further, the small tubular or muffle furnaces used in most scientific studies [18,19] can lead to an inhomogeneous transfer of heat and reaction gases toward the activated carbon bedding because this is not moved as in a rotary kiln or fluidized bed oven. Hence, a transfer of the results to technical-scale furnaces is often impossible.

As systematic and quantitative information on the thermal reactivation of spent AC is still missing in the literature, the main objective of the current study was the establishment of systematic and quantitative information regarding the thermal reactivation of activated carbon under near-practical conditions. The large dataset and the use of statistical methods for the design of the experiments allowed for an advanced evaluation of the results and the development of predictive models. This is, to our knowledge, the first time that a statistically based predictive model for the thermal reactivation of ACs has been published. Whereas the influence of the reactivation conditions on the physicochemical properties of the AC was published elsewhere [12], the current paper is focused on the statistical modeling approach and the adsorptive properties.

## 2. Materials and Methods

### 2.1. Preparation of Exhausted AC

A fossil-coal-based exhausted AC used for the removal of personal care and pharmaceutical products and other trace organics from wastewater was obtained in wet conditions from a municipal sewage treatment facility in Germany. Prior to the reactivation experiments, the exhausted AC was washed with water five times to remove coarse impurities (plastics, mud, and shells), dried in an air oven (Universal Oven Model UNB 500, Memmert GmbH, Schwabach, Germany) at 105 °C for 24 h, and stored in an airtight container until further usage.

### 2.2. Reactivation Conditions and Target Parameters

The influence of the reactivation conditions heating rate, heating time, $H_2O/N_2$ volume ratio, and $CO_2/N_2$ volume ratio on the specific surface area (expressed as BET surface area), energy consumption, yield, and adsorption capacity of RAC was investigated.

### 2.3. Thermal Reactivation Protocol

A custom-made bench-scale rotary kiln furnace (see Figure S1) was employed for conducting the thermal reactivation experiments. $N_2$ gas with a flow rate of 250 L/h was constantly supplied throughout the experiment. Furthermore, 400 g of the granular exhausted AC was placed in a rotating stainless steel reactor and heated to the desired temperature at a specified heating rate and heating time. After the reactivation, the RAC was cooled down to room temperature under a pure nitrogen atmosphere. Before setting the test parameters, the furnace was first heated to 300 °C in each experiment. Subsequently, a heating time from 40 to 80 min and a heating rate from 250 to 400 K/min were set, resulting in a maximum temperature between 450 °C and 900 °C. It is likely that below 450 °C, the reactivation success will be too low, and, above 900 °C, the pore system might collapse, or too high a mass loss might occur. Furthermore, after each reactivation experiment, the energy consumption and RAC yield were noted and the adsorption capacity for diatrizoic acid was determined. The energy consumption was always determined for the whole experiment, i.e., the recorded values refer to an original mass of 400 g exhausted carbon. Including preheating and cooling to room temperature, each reactivation experiment took about 24 h.

### 2.4. Determination of RAC's Adsorption Capacity

Prior to the adsorption experiment, the particle size of the RACs was brought down to below 45 μm using a ball mill (Pulverisette 7, Fritsch gmbH, Idar-Oberstein, Germany). In this work, diatrizoic acid, a contrast agent applied in X-ray diagnostics, was used as an example for compounds that cannot be efficiently removed in biological wastewater treatment. The diatrizoic acid was procured from Sigma-Aldrich, Taufkirchen, Germany. The diatrizoic acid adsorption capacity was determined using so-called "bottle-point isotherms" [20]. All adsorption experiments were carried out in a synthetic model water (0.5 mmol/L $NaHCO_3$, 0.3 mmol/L $CaCl_2$, and 0.2 mmol/L $MgSO_4$) made from ultrapure water as described by Landwehrkamp et al. [17]. The pH value of the synthetic model water was ~7.4 and the conductivity ~200 μs/cm at 25 °C. In a typical batch experiment, different amounts (5–50 mg/L) of RAC were added to the flasks containing diatrizoic acid (3 mg/L) solution. It should be noted that at the pH value of the experiments, the diatrizoic acid (pKa-value ~3.4) was almost exclusively present as the corresponding anion, i.e., diatrizoate. The resulting suspension was shaken at 150 rpm at room temperature using a mechanical shaker (Laboshake, Gerhardt GmbH & Co. KG, Königswinter, Germany). After 24 h contact time, the suspension was filtered through 0.45 μm cellulose nitrate filters (Ahlstrom GmbH, Bärenstein, Germany) and the residual diatrizoic acid concentration was analyzed using a Lambda 20 UV spectrophotometer (Perkin Elmer, Rodgau, Germany). The diatrozoic acid concentration at equilibrium varied between 0.1 mg/L and 3 mg/L. All experiments were repeated at least two times.

The solid phase loading (q in $\frac{g}{kg}$) of the RACs was determined using Equation (1):

$$q = (c_0 - c_e)\frac{V}{M} * 1000 \tag{1}$$

where $c_0$ and $c_e$ are the initial and equilibrium concentrations of the diatrizoic acid (mg/L), V is the volume of solution (L), and M is the mass of the RAC (g).

Lineweaver–Burk linearization of the Langmuir model (2) and a simple linearization form of the Freundlich model (3) were used to fit the equilibrium data [21]:

$$\frac{1}{q_e} = \left(\frac{1}{q_{max}K_L}\right)\frac{1}{c_e} + \frac{1}{q_{max}} \tag{2}$$

$$\log\left(\frac{q_e}{\frac{g}{kg}}\right) = \log\left(\frac{K_F}{\frac{\left(\frac{mg}{g}\right)}{\left(\frac{mg}{L}\right)^n}}\right) + \frac{1}{n}\log\left(\frac{c_e}{\frac{mg}{L}}\right) \tag{3}$$

where $q_e$ (g/kg) is the capacity of adsorbate uptake of diatrizoic acid at equilibrium and $q_{max}$ (g/kg) the maximum saturated monolayer adsorption capacity, $K_L$ (L/(mg)) the Langmuir isotherm constant related to the affinity between the respective AC and the dosed diatrizoic acid, $K_F$ $(mg/g)/(mg/L)^n$ the Freundlich constant, and n (dimensionless) the Freundlich intensity parameter, which indicates the magnitude of the adsorption driving force or the surface heterogeneity. Since the argument of the logarithm must be unitless, the corresponding parameters are divided by their units.

### 2.5. Design of Experiments

The so-called response surface methodology (RSM) is a statistical tool for the assessment of the influence of individual experimental factors (or their combinations) on product properties [22,23]. Besides a fundamental understanding of the process itself, the statistical model allows predictions, i.e., it is possible to model the value of the response factors (R) for a defined set of experimental parameters.

Often, the quantitative dependence of one (or more) target variable(s) on a few factors is to be determined in detail. Hence, it is often impossible to apply a linear fit. The

nonlinearity is crucial when the position of a maximum (e.g., the BET surface) or a minimum (e.g., the energy consumption) is being searched for. In most cases, a quadratic model is then used to empirically describe the dependence of the target variable R on the factors $X_i$:

$$R = \beta_0 + \sum_{i=1}^{n} \beta_i X_i + \sum_{i=1}^{n} \beta_{ii} X_i^2 + \sum_{i=1}^{n-1} \sum_{j=i+1}^{n} \beta_{ij} X_i X_j + \varepsilon \tag{4}$$

where $R$ is the predicted response factor, $\beta_0$ is the constant coefficient, $\beta_{ii}$ is the quadratic coefficients, $\beta_{ij}$ is the interaction coefficients, $X_i, X_j$ are the coded values of the factors considered, and $\varepsilon$ is the random deviation observed in the response $R$. $X_i, X_j$ and the respective products of both are the so-called predictors.

In this study, the following responses (dimensionless) were considered and determined: $R_1$ = BET surface area/$(m^2/g)$, $R_2$ = energy consumption/$(kWh)$, $R_3$ = yield/$(\%)$, and $R_4$ = diatrizoic acid adsorption capacity/$(g/kg)$. The diatrizoic acid adsorption capacity ($R_4$) was determined from the parameters of the Langmuir and Freundlich isotherms for the uptake of diatrizoic acid from water (see Table S1). Based on the higher correlation coefficient value ($R^2$), the Langmuir isotherm fitted well to the adsorption data. Therefore, the determined adsorption capacity ($R_4$) provided in Table 1 is the (theoretical) maximum solid phase concentration $q_m$ calculated from the Lineweaver–Burke linearization (Equation (2)) of the Langmuir equation.

**Table 1.** Experimental design matrix and determined responses for each reactivation process. * For comparison: the BET surface area of the respective fresh AC was measured to be 843 $m^2/g$, adsorption capacity was 59.6 g/kg.

| | Run | $X_1$ | $X_2$ | $X_3$ | $X_4$ | $R_1$ * | $R_2$ | $R_3$ | $R_4$ * |
|---|---|---|---|---|---|---|---|---|---|
| | | Heating Rate in K/h | Heating Time in min | $H_2O/N_2$ Volume Ratio | $CO_2/N_2$ Volume Ratio | BET Surface in $m^2/g$ | Energy Consumption in kWh | Yield in % | Diatrizoic Acid Adsorption Capacity in g/kg |
| Factorial points | 1 | 225 | 40 | 0 | 0 | 590 | 2.5 | 92.1 | 23.4 |
| | 2 | 450 | 40 | 0 | 0 | 673 | 3.2 | 90.3 | 42.7 |
| | 3 | 225 | 80 | 0 | 0 | 635 | 3.8 | 90.0 | 40.8 |
| | 4 | 450 | 80 | 0 | 0 | 681 | 6 | 87.5 | 47.1 |
| | 5 | 225 | 40 | 0.3 | 0 | 597 | 2.5 | 91.5 | 34.1 |
| | 6 | 450 | 40 | 0.3 | 0 | 687 | 3.3 | 89.8 | 60 |
| | 7 | 225 | 80 | 0.3 | 0 | 659 | 3.9 | 89.8 | 35.4 |
| | 8 | 450 | 80 | 0.3 | 0 | 762 | 6 | 85.8 | 62.4 |
| | 9 | 225 | 40 | 0 | 0.3 | 598 | 2.4 | 92.0 | 30.3 |
| | 10 | 450 | 40 | 0 | 0.3 | 664 | 3.4 | 90.5 | 41.3 |
| | 11 | 225 | 80 | 0 | 0.3 | 649 | 3.8 | 90.4 | 47.7 |
| | 12 | 450 | 80 | 0 | 0.3 | 735 | 5.9 | 86.4 | 51.5 |
| | 13 | 225 | 40 | 0.3 | 0.3 | 644 | 2.6 | 91.4 | 36.2 |
| | 14 | 450 | 40 | 0.3 | 0.3 | 653 | 3.4 | 89.5 | 58.6 |
| | 15 | 225 | 80 | 0.3 | 0.3 | 633 | 4 | 89.9 | 50.6 |
| | 16 | 450 | 80 | 0.3 | 0.3 | 769 | 6 | 84.9 | 69.7 |
| Axial points | 17 | 157 | 60 | 0.15 | 0.15 | 635 | 2.8 | 91.3 | 29.2 |
| | 18 | 518 | 60 | 0.15 | 0.15 | 728 | 4.9 | 87.5 | 65 |
| | 19 | 338 | 28 | 0.15 | 0.15 | 668 | 2.3 | 91.7 | 32 |
| | 20 | 338 | 92 | 0.15 | 0.15 | 737 | 5.6 | 87.3 | 58.6 |
| | 21 | 338 | 60 | 0.4 | 0.15 | 686 | 3.8 | 89.4 | 52.1 |
| | 22 | 338 | 60 | 0.15 | 0.4 | 699 | 3.8 | 89.5 | 53.1 |
| Center point | 23 | 338 | 60 | 0.15 | 0.15 | 680 | 3.7 | 89.9 | 38.8 |
| | 24 | 338 | 60 | 0.15 | 0.15 | 690 | 3.8 | 89.3 | 39.7 |
| | 25 | 338 | 60 | 0.15 | 0.15 | 699 | 3.8 | 87.9 | 41 |
| | 26 | 338 | 60 | 0.15 | 0.15 | 674 | 3.9 | 89.5 | 43.3 |

Four reactivation conditions (factors) were investigated at two stages each, namely heating rate (225 and 450 °C/h), heating time (40 and 80 min), $H_2O/N_2$ volume ratio (0 and 0.3), and $CO_2/N_2$ volume ratio (0 and 0.3), respectively. For reasons of comparison, all responses were also determined for the respective fresh AC and the exhausted AC sample.

To adequately represent the quadratic effects, an experimental design with more than two stages is necessary. However, since a full factorial experimental design with three stages would result in unreasonably high experimental effort, a central composite design (CCD) was chosen here. Accordingly, a $2^4$ full factorial experimental design with an additional center point and additional $2 \times 4 = 8$ star points was designed [24]. A centrally composed experimental design is orthogonal with respect to all terms in the model (Equation (4)) if

$$\alpha^2 = \frac{1}{2}\left(\sqrt{N \cdot N_s} - N_s\right) \tag{5}$$

where $\alpha$ is the distance (in coded units) of each star point from the central point, $N_s$ is the number of single experiments for the $2^4$ full factorial experimental design without a center point (here: $N_s = 2^4 = 16$), and N is the total number of experiments for the CCD (here: $N_s = 28$, considering the eight star points and if a fourfold repetition of the test is carried out at the central point). The orthogonal experimental design is of benefit for the mathematical evaluation since:

- the estimates for the coefficients β in model (Equation (1)) are independent of each other (i.e., they do not influence each other), and
- for given step values for $-1$ and 1 (code values) and a given number of individual trials, one obtains the narrowest possible confidence intervals for the coefficients β [24].

The determined coded $\alpha$ values of the individual factors used in the experimental design and their corresponding non-coded values are shown in Table 2.

**Table 2.** Experimental range and levels of the independent test factors.

| Factors | Label | $\alpha = -1.61$ | $-1$ | $0$ | $1$ | $1.61$ |
|---|---|---|---|---|---|---|
| Heating rate (K/h) | $X_1$ | 157 | 225 | 338 | 450 | 518 |
| Heating time (min) | $X_2$ | 28 | 40 | 60 | 80 | 92 |
| $H_2O/N_2$ volume ratio | $X_3$ | $-0.15$ | 0 | 0.15 | 0.3 | 0.4 |
| $CO_2/N_2$ volume ratio | $X_4$ | $-0.15$ | 0 | 0.15 | 0.3 | 0.4 |

The Minitab (version 19) statistical software was applied for experimental design, for the regression and graphical analyses of the obtained data, and for the forecasts in the reactivation process. The analysis of variance used in Minitab and other important terminology are explained in detail in Supplementary Information (S1). Given the fact that two experiments with negative $\alpha$ values for the $H_2O/N_2$ and $CO_2/N_2$ volume ratio were evidently not possible, D-optimization was performed using Minitab to improve the initial experimental design. D-optimization allows limitations in the experimental design to be taken into account, i.e., combinations that are not technically or physically possible [24]. This has resulted in 26 reactivation experiments instead of 28 (see Table 1). All reactivation experiments were conducted in a random sequence as proposed by Minitab.

## 3. Results and Discussion

### 3.1. Thermal Reactivation

For each reactivation experiment, the corresponding responses for BET surface area ($R_1$), energy consumption per reactivation experiment ($R_2$), yield ($R_3$), and diatrizoic acid adsorption capacity ($R_4$) were determined. The results are shown in Table 1. While the experimentally determined BET surface ($R_1$) of the RACs ranged between 590 and 769, the energy consumption ($R_2$), yield ($R_3$), and adsorption capacity ($R_4$) ranged between 2.3 and

6, 84.9 and 92.1, and 23.4 and 69.7, respectively. Furthermore, the repetitions of the center point resulted in minor deviations, underlining the good reproducibility of the experiments.

### 3.2. Statistical Model Analysis

A mathematical quadratic polynomial regression expression (Equation (4)) was used to analyze the influence of factors on the responses using Minitab. The significance of the factors was assessed using a standard analysis of variance (ANOVA [25]) with $\alpha$ = 0.05 as the significance level. All the results of ANOVA are listed in Supplementary Information (Tables S2–S5). The respective full quadratic equations with the exclusion of insignificant factors for the responses BET surface ($R_1$), energy consumption ($R_2$), yield ($R_3$), and adsorption capacity ($R_4$) are given in Equations (6)–(9), respectively.

$$R_1 = 470.5 + 0.3232X_1 + 1.248X_2 + 329X_3 - 793X_3^2 \tag{6}$$

$$R_2 = 2.13 - 0.00217X_1 - 0.0171X_2 + 0.00017X_2^2 + 0.00014X_1 \cdot X_2 \tag{7}$$

$$R_3 = 93.22 + 0.00221X_1 + 0.0102X_2 - 2.351X_3 - 0.000236X_1 \cdot X_2 \tag{8}$$

$$R_4 = 2.83 + 0.0508X_1 + 0.2869X_2 - 34.0X_3 + 18.97X_4 + 0.2001X_1 \cdot X_3 \tag{9}$$

A positive sign in front of the terms indicates a synergistic effect, whereas a negative sign indicates an antagonistic effect. The impact of the impact factors on the responses are visualized using Pareto charts and two-dimensional contour plots determined using Minitab. In the Pareto chart, the absolute values of the standardized effects are shown ordered from the largest to the smallest effect. The standardized effects are t-statistics used to test the null hypothesis that the effect is equal to 0. In this study, all standardized effects correspond to the absolute value of the t-statistic for the respective coefficient of the predictors. In addition, a reference line is displayed in the diagram to show which effects are statistically significant. The reference line for statistical significance is the critical t-value which depends on the chosen significance level (here $\alpha$ = 0.05). It equals the $(1 - \alpha/2)$th quantile of a t-distribution with the degrees of freedom corresponding to the degrees of freedom of the error term (number of reactivation experiments—number of predictors − 1). If the standardized effect is larger than the critical t-value, the null hypothesis can be rejected, and the effect is statistically significant. For further information, see [26]. Only significant standardized effects are shown in the following Pareto charts.

It is evidenced from the Pareto chart analysis (Figure 1a) that the BET surface ($R_1$) is influenced significantly by the linear terms heating rate ($X_1$), heating time ($X_2$), and $H_2O/N_2$ volume ratio $X_3$, and the quadratic term $(X_3)^2$. Figure 1b–d show the conjugate effect of the heating rate ($X_1$) and heating time ($X_2$) on the BET surface ($R_1$) in which the $H_2O/N_2$ volume ratio ($X_3$) varies between 0, 0.2, and 0.4 (expressed in the figure as "set value"). It can be seen from Figure 1b that a higher heating rate ($X_1$) and heating time ($X_2$) can certainly enhance the BET surface ($R_1$), which might be attributed to the better mobilization of previously adsorbed compounds from the AC matrix. Equally, a $H_2O/N_2$ ratio ($X_3$) increasing from 0 to 0.2 (Figure 1c) contributes to the BET surface ($R_1$) and a further increase in the $H_2O/N_2$ ratio ($X_3$) to 0.4 (Figure 1d) results in a decrease in the BET surface ($R_1$). A possible explanation for this behavior might be that water vapor reacts and oxidizes the residual adsorbed compounds and removes them from the material [27]. This helps in the restoration of the original pore structure of the AC. However, a higher water vapor concentration can lead to a reduction in the BET surface ($R_1$), as pore widening is more pronounced and small pores combine to form a larger pore (over-reactivation). Furthermore, squared $X_3$ interactions showed almost no influence on the BET surface ($R_1$), whereas increasing or decreasing the $CO_2/N_2$ ratio ($X_4$) showed no effect on $R_1$.

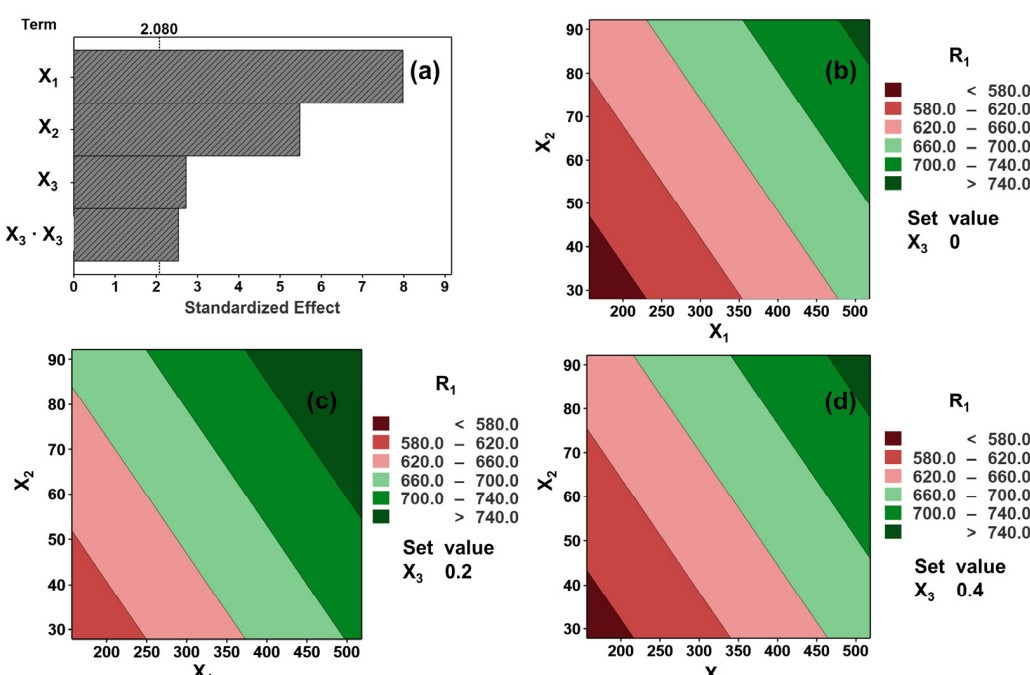

**Figure 1.** (**a**) Pareto chart of the influence of factors on BET surface area ($R_1$) and (**b–d**) contour plots of BET surface area ($R_1$) showing the influence of heating rate ($X_1$) and heating time ($X_2$) at different $H_2O/N_2$ ratio ($X_3$) set values.

From an economic point of view, the energy consumption ($R_2$) is considered one of the vital parameters for the reactivation processes. The results reveal that the model and the terms heating time ($X_2$) and heating rate ($X_1$) are significant to the response. The energy consumption ($R_2$) is greatly affected by the linear terms heating time ($X_2$) and heating rate ($X_1$), by the interactive term $X_1 \cdot X_2$, and the quadratic term $(X_2)^2$ (see Figure 2a). It is evident from Figure 2b that energy consumption ($R_2$) increases with an increase in heating time ($X_2$) and heating rate ($X_1$).

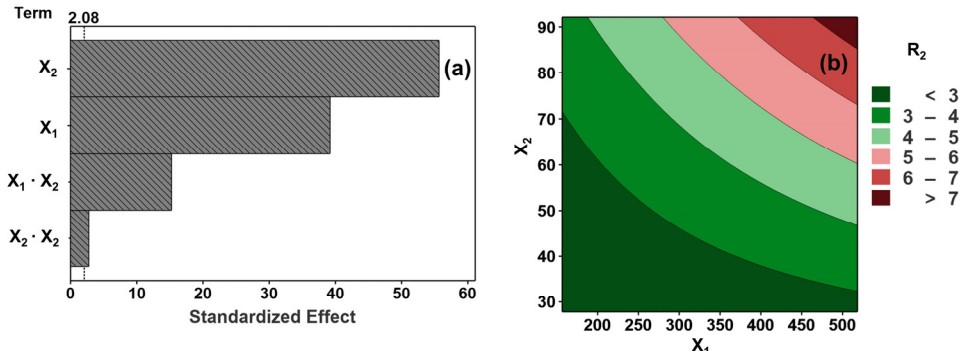

**Figure 2.** (**a**) Pareto chart of the influence of factors on energy consumption ($R_2$) and (**b**) the contour plot of energy consumption ($R_2$) showing the influence of heating rate ($X_1$) and heating time ($X_2$).

The yield (see Figure 3a) ($R_3$) is impacted strongly by the heating rate ($X_1$) and heating time ($X_2$), followed by their interactive terms ($X_1 \cdot X_2$), and the $H_2O/N_2$ ratio ($X_3$). Figure 3b–d show the contour plots of the effects of heating rate ($X_1$) and heating time ($X_2$) on the yield ($R_3$) by varying the $H_2O/N_2$ ratio ($X_3$). It is shown that increasing the heating rate ($X_1$) and heating time ($X_2$) results in a decreased yield ($R_3$). This was expected, as heating in the presence of oxidizing gas leads to a burn-off of previously adsorbed compounds (the desired effect) and carbon material itself (unwanted effect). $CO_2$ has a certain oxidation potential as described using the Boudouard reaction [28]; however, water vapor is generally regarded

as a stronger oxidation agent in activated carbon activation [29]. This was confirmed by the results obtained within this study.

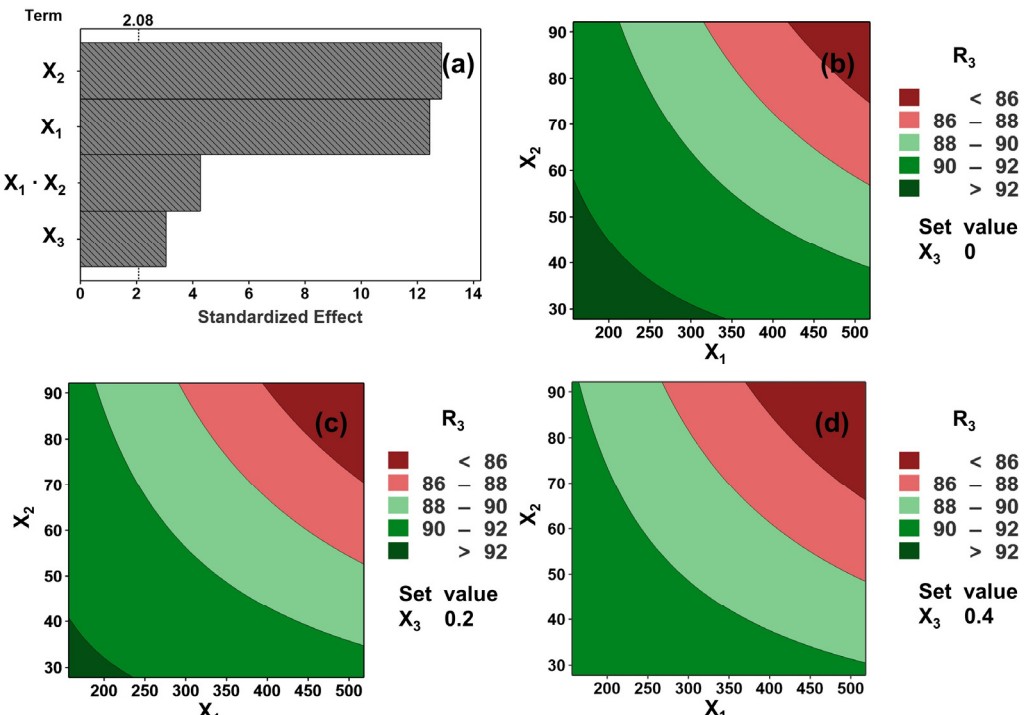

**Figure 3.** (**a**) Pareto chart of the influence of factors on yield ($R_3$) and (**b–d**) contour plots of yield ($R_3$) showing the influence of heating rate ($X_1$) and heating time ($X_2$) at different $H_2O/N_2$ ratio ($X_3$) set values.

It is observed from Figure 4a that the adsorption capacity ($R_4$) is strongly dependent on the heating rate ($X_1$) term followed by the terms heating time ($X_2$), $H_2O/N_2$ ratio ($X_3$), $X_1*X_3$, and $CO_2/N_2$ ratio ($X_4$). The influence of the $CO_2/N_2$ ratio ($X_4$) on the response is only seen for adsorption capacity ($R_4$), but not for the other responses. It is apparent from Figure 4b–d that increasing the heating time ($X_2$) and heating rate ($X_1$) leads to an increase in the diatrizoic acid adsorption capacity. This is most likely because the elimination of strongly or chemically adsorbed compounds from the AC matrix is quite difficult at a lower heating time ($X_2$) and heating rate ($X_1$) and, hence, the restoration of the original pore structure fails. Furthermore, increasing both the $H_2O/N_2$ ratio ($X_3$) and $CO_2/N_2$ ratio ($X_4$) from 0 to 0.4 has promoted the diatrizoic acid adsorption capacity of the RACs. Interestingly, the authors found in their study of the influence of the reactivation conditions on the physio-chemical properties of the RACs [12] that the surface chemistry (number and nature of functional groups) changes differently also depending on the $H_2O/N_2$ ratio ($X_3$) and $CO_2/N_2$ ratio ($X_4$). This certainly can impact the adsorption capacity of the AC for diatrizoic acid.

The suitability of the developed polynomial regression equation was further assessed by determining the coefficient of determination $R^2$. This was evaluated by plotting the experimental values of the single response factors (i.e., the real values that were obtained within the reactivation experiments) against the values calculated using Equations (6)–(9) (cf. Figure 5). The determined $R^2$ values for the developed polynomial models are 83.3% for the BET surface ($R_1$), 99.6% for the energy consumption ($R_2$), 94.3% for the yield ($R_3$), and 89.7% for the adsorption capacity ($R_4$), which implies that only 16.7%, 0.4%, 5.7%, and 10.3% of the total variation cannot be explained by the respective model and underlines the reliability of the developed models.

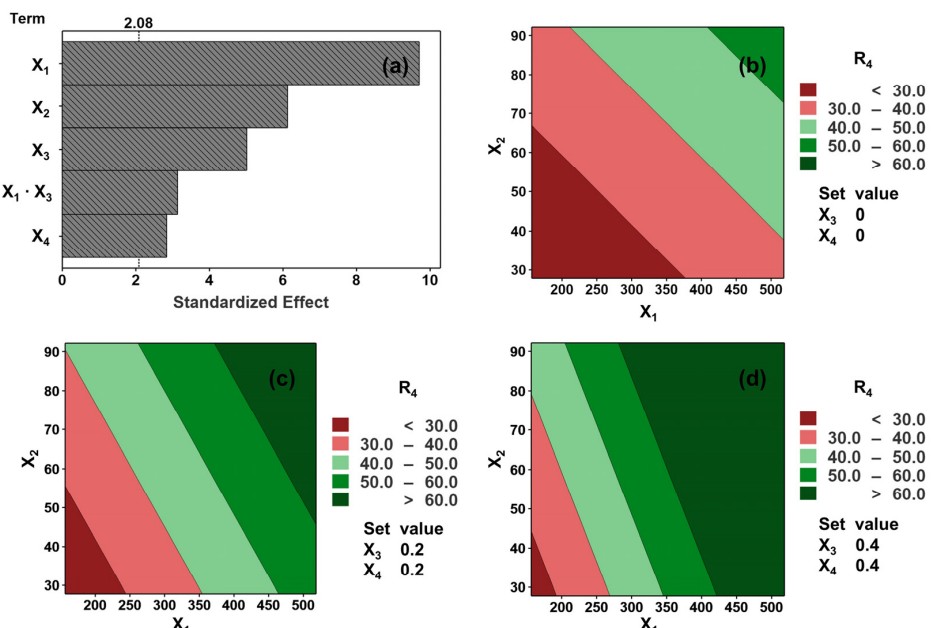

**Figure 4.** (**a**) Pareto chart of the influence of factors on the maximum adsorption capacity (R$_4$) and (**b**–**d**) contour plots of adsorption capacity (R$_4$) showing the influence of heating rate (X$_1$) and heating time (X$_2$) at different H$_2$O/N$_2$ ratio (X$_3$) and CO$_2$/N$_2$ ratio (X$_4$) set values.

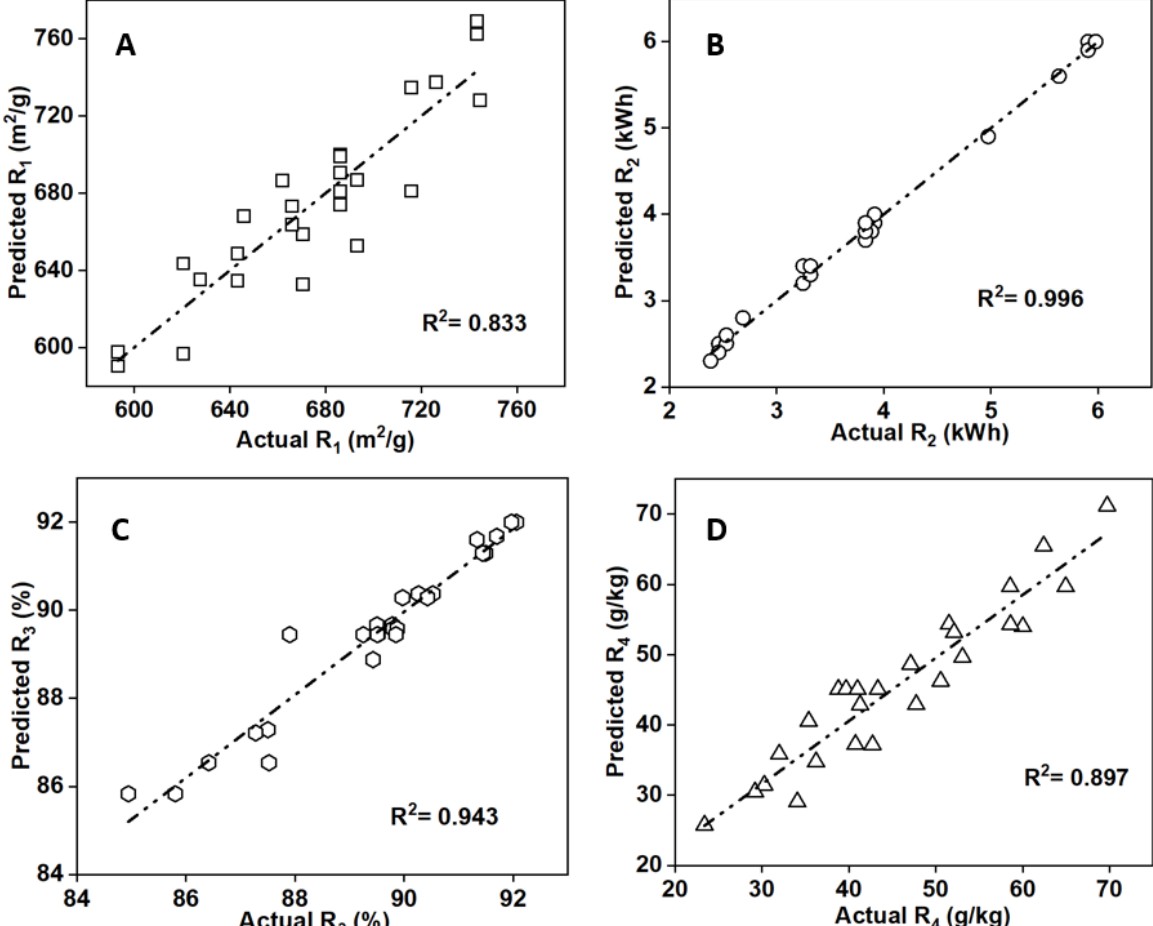

**Figure 5.** Predicted vs. actual values of (**A**) BET surface area (R$_1$), (**B**) energy consumption (R$_2$), (**C**) yield (R$_3$), and (**D**) diatrizoic acid adsorption capacity (R$_4$).

*3.3. Production of Reactivates with a Defined BET Surface Area at Lowest Energy Consumption*

The BET surface area ($R_1$) is one of the most important parameters for RACs. The desired or necessary BET surface of RAC is strongly dependent on the respective application (e.g., drinking water treatment, wastewater treatment, treatment of landfill leachates, etc.), whereas applications for emission control typically require RAC of a lower quality than drinking water treatment. Providing RACs with higher BET surface areas is connected to higher energy consumption for the reactivation and thus of disadvantages for reactivating companies, also against the background that a higher BET surface area does not necessarily imply a higher adsorption capacity. Thus, the target of this optimization was to predict the optimal reactivation conditions for the production of a RAC with a BET surface area of 630 $m^2/g$ with the additional goal of achieving it with the minimum energy consumption. The value of 630 $m^2/g$ was chosen to make the modeling results comparable with some of the experimental results (runs 3, 15, and 17).

The proposed reactivation conditions extracted from the model are given in Table 3. For these conditions, an energy consumption of 2.5 kWh was predicted. Please note that the $CO_2/N_2$ ratio is not part of the model and that the model proposes no addition of $H_2O$ to realize the target. The parameter settings proposed by the model were used to produce a sample "OC" (Table 4). The BET surface area of the produced RAC was determined as 612 $m^2/g$, whereas the energy consumed was 2.4 kWh and the yield obtained was 91%. The obtained BET surface area value was within a 95% confidence interval of the predicted BET surface area value. Furthermore, the predicted and the actual energy consumption did not differ within the given measurement precision ($\pm 0.1$ kWh), thereby underlining the quality of the model.

**Table 3.** Optimized conditions proposed by the model for the targeted BET surface area under minimum energy demand.

| Proposed Conditions | | | BET-Surface Area ($m^2/g$) | | Energy Consumption (kWh) | |
|---|---|---|---|---|---|---|
| Heating rate (K/h) | Heating time (min) | $H_2O/N_2$ ratio | Model prediction | Exp. result | Model prediction | Exp. result |
| 387 | 28 | 0 | 630 | 612 | 2.5 | 2.4 |

**Table 4.** Adjusted parameters and results of experimental runs 3, 15, 17 and with optimized conditions (OC).

| Run | Heating Rate (K/h) | Heating Time (min) | $H_2O/N_2$ Ratio | BET-Surface Area ($m^2/g$) | Energy Consumption (kWh) |
|---|---|---|---|---|---|
| 3 | 225 | 80 | 0 | 635 | 3.8 |
| 15 | 225 | 80 | 0.3 | 633 | 4 |
| 17 | 157 | 60 | 0.15 | 635 | 2.8 |
| OC | 387 | 28 | 0 | 612 | 2.4 |

A comparison with the results of runs 3, 15, and 17 (Table 4) shows that in case of non-optimal reactivation conditions, the energy consumption will be much larger for the same targeted BET surface area (up to about 65% related to the optimum).

## 4. Conclusions and Outlook

A simple and robust method using a central composite design of experiments and response surface methodology with nonlinear regression was used to develop a predictive model for the reactivation of the exhausted ACs used in water treatment. The model is able to describe the influence of the reactivation conditions on several product quality and process criteria. By means of the model, the simultaneous optimization of several key

performance parameters can be carried out by predicting suitable reactivation conditions. The influence of the reactivation conditions heating rate, heating time, $H_2O/N_2$-volume ratio, and $CO_2/N_2$-volume ratio on the specific surface area (expressed as BET surface area), energy consumption, yield, and diatrizoic acid adsorption capacity was investigated. For better scalability of the laboratory results to the technical scale, a bench-scale rotary kiln was used, allowing the reactivation of 400 g of sample per batch. It was observed that:

- heating rate and heating time had the greatest influence on all the considered process and product criteria.
- the $H_2O/N_2$ ratio generally showed less effect, but a proper adjustment can help to increase the yield or the adsorption capacity.
- the $CO_2/N_2$ ratio only affected the adsorption capacity for diatrizoic acid.
- The results further show that the correlations between the effects (reactivation conditions or their combinations) have considerably nonlinear components. Therefore, the conducted nonlinear regression was necessary for the statistical modeling.

The proposed methodology is a powerful tool to optimize the thermal reactivation process of a specific AC and to adjust the operating parameters of reactivation kilns in such a way that, for example:

1. a recovery of the adsorption properties as complete as possible can be achieved at a minimum mass loss,
2. a tailor-made reactivate with desired characteristics can be produced, and
3. the energy input and $CO_2$ emissions can be minimized to produce reactivates with sufficient quality for less demanding treatment processes (downscaling).

A future application of the methodology could be a lifecycle analysis in which the reactivation process and the filter reactivation frequency in a treatment process are matched to reduce the overall energy consumption. Lower quality of the RAC due to an energy-optimized reactivation can result in a shorter lifetime of the filter and thus, a higher reactivation frequency. The shorter lifespan of the filter, however, could be overcompensated for by the lower energy consumption for the less extensive reactivation, resulting in a net energy saving. Vilen et al. [30] postulated in their comparative lifecycle analysis of activated carbons that the majority of the $CO_2$ footprint of reactivated carbons stems from the make-up (fresh) carbon that is necessary to compensate for the mass loss during reactivation. The proposed methodology can help to optimize reactivation processes in terms of yield and thus contribute to a more environmentally friendly activated carbon application.

**Supplementary Materials:** The following supporting information can be downloaded at: https://www.mdpi.com/article/10.3390/c9040115/s1 Figure S1: Photo image of a bench-scale rotary kiln furnace set-up used for the thermal reactivation process, Table S1: Calculated isotherm parameters for DA adsorption onto RACs and fresh AC, Table S2: Analysis of Variance for R1, Table S3: Analysis of Variance for R2, Table S4: Analysis of Variance for R3, Table S5: Analysis of Variance for R4, S1: Analysis of Variance using the F-Test of the overall significance, S2: Standardized Effects.

**Author Contributions:** Conceptualization, K.R. and L.L.; methodology, K.R.; software, K.R.; validation, K.R., S.P. and L.L.; investigation, K.R., L.L. and V.M.; writing—original draft preparation, K.R.; writing—review and editing, S.P., L.L., D.B., C.P. and C.B.; visualization, K.R. and L.L.; supervision, D.B., C.P. and S.P.; funding acquisition D.B., S.P., L.L. and C.B. All authors have read and agreed to the published version of the manuscript.

**Funding:** The Chairs for Mechanical Process Engineering/Water Technology and Thermal Process Engineering would like to thank the German Federation of Industrial Research Associations within the AiF project 20617 N for the financial support and Donau Carbon for providing the activated carbon. The publication costs were covered by the Open Access Publication Fund of the University of Duisburg-Essen.

**Data Availability Statement:** Data are provided in the Supplementary Information.

**Conflicts of Interest:** The authors declare no conflict of interest.

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
