# Peer review of "Eliminating Luck and Chance in the Reactivation Process: A Systematic and Quantitative Study of the Thermal Reactivation of Activated Carbons"

_carbon_

Round 1
Reviewer 1 Report
Comments and Suggestions for Authors
This is a valuable study on the optimization of activated carbon reactivation conditions and the development of a model for a more targeted process optimization. However, the manuscript needs some improvement in order to improve clarity and make it easier for the reader to follow the discussion.
The meaning of the parameters X_iand R_i are introduced at the beginning and the parameters afterwards used consistently. However, this makes the manuscript hard to follow for the reader. It is impossible to remember the meaning of the various index numbers. Some more intuitive indices would help a lot.
It remained unclear to me which adsorption parameter the authors used when mentioning ‘adsorption capacity’. This should be written in a clearer way.
line 80: Is indeed highest maximum capacity meant here or rather capacity in terms of removal degree. Please refine.
line 182; If simple capacity is meant here than conditions of adsorbate/adsorbent dosage need to be mentioned.
Properties of the virgin carbon (BET area, ash and carbon content, PZC,…) should be described in the manuscript.
What is the pH in the bath experiments? Please report. Different reactivation procedures for AC can result in different surface chemistry of the product and thus the pH of the suspensions can differ if it is not adjusted to a fixed value. Please discuss whether this has played a role in the experiments.
What is the pKa value of diatrizoic acid and what is the speciation of the compound in the adsorption tests?
Do the authors assume that the key findings on optimal parameters for reactivation are specific for the type of carbon and contaminant tested here or do they consider them to be widely applicable? This didn’t become very clear so far in the manuscript.
Why was the AC used in this particle size range for regeneration? What is done in large-scale reactivation, is the granulation of the AC broken to powder for regeneration and is the carbon granulated afterwards again? What would happen if granules are directly reactivated?
Freundlich and Langmuir isotherms should be plotted in figures and not just isotherm parameters reported in a table. This is also not correct without additional information as Freundlich isotherm parameters are valid only for the concentration ranges in which they were measured.
Table 1: Heating rate should be K/min
Table 3 and related text in Chapter 3.3: Energy consumption should be reported per mass of AC regenerated.
Line 358: It should be specific surface area not inner surface area.
Fenton-driven regeneration of AC could be mentioned in the discussion of alternative regeneration options in the introduction with a highly cited paper in this field (https://doi.org/10.1016/j.apcatb.2009.02.002). In the same way, there is a recent review on regeneration of carbon-based adsorbents by electric potentials (https://doi.org/10.1016/j.cej.2023.144354).
Reviewer 2 Report
Comments and Suggestions for Authors
The aim of the manuscript entitled “Eliminating luck and chance in the reactivation process: a systematic and quantitive study of the thermal reactivation of activated carbons” is to address the growing demand for activated carbon while mitigating its environmentally taxing production by systematically and quantitatively studying the thermal reactivation process. Despite the significant potential of activated carbon reactivation in reducing the need for fresh material, existing knowledge about this process remains largely empirical. This study aims to develop a predictive model for reactivated carbon production, focusing on product quality and energy efficiency. Through a series of carefully designed experiments and statistical analysis, the research investigates the impact of reactivation conditions on critical parameters such as inner surface area, energy consumption, yield, and adsorption capacity. Ultimately, the goal is to optimize the reactivation process, resulting in substantial energy savings and reactivated carbon properties closely aligned with predictions. The manuscript is interesting and fits well with the scope of the journal. It was very well prepared, and I only have a few minor comments.
While the abstract mentions that the study focuses on the influence of reactivation conditions on various parameters, it doesn't provide specific findings or results. Including some key findings or outcomes in the abstract would enhance its informativeness.
The abstract briefly mentions the environmental concerns and demand for activated carbon but could benefit from a more detailed explanation of the significance of the research. Why is it important to develop a predictive model for reactivated carbon production? How does this address pressing environmental issues?
The introduction is informative, but the motivation for the paper and the novelty should be clearly stated in its final paragraph.
Materials and methods are given in detail.
The font in the Tables differs from the one in the main text. The results and discussion are well presented overall.
Some sentences in the conclusion are quite long and complex. It's important to maintain clarity and conciseness in scientific writing. Breaking down complex ideas into shorter, more digestible sentences would improve readability. Also, the mention of future applications and life-cycle analysis is positive, but it would be helpful to explain the relevance of these potential applications more explicitly. How do they connect to the current research, and what broader impact could they have?
Comments on the Quality of English LanguageMinor changes are required.
Round 2
Reviewer 1 Report
Comments and Suggestions for Authors
The improvements done by the author are acknowledged. I recommend acceptance.
Reviewer 2 Report
Comments and Suggestions for Authors
The authors addressed all my comments.
Comments on the Quality of English LanguageMinor changes are required.